# Refractory Age-Related Macular Degeneration Due to Concurrent Central Serous Chorioretinopathy in Previously Well-Controlled Eyes

**DOI:** 10.3390/ph16010089

**Published:** 2023-01-08

**Authors:** Chikako Hara, Taku Wakabayashi, Kaori Sayanagi, Kohji Nishida

**Affiliations:** 1Department of Ophthalmology, Osaka University Graduate School of Medicine, Suita 5650871, Japan; 2Wills Eye Hospital, Mid Atlantic Retina, Thomas Jefferson University, Philadelphia, PA 19107, USA

**Keywords:** aflibercept, age-related macular degeneration, anti-vascular endothelial growth factor drug, central serous chorioretinopathy, tachyphylaxis

## Abstract

**Background**: During the treatment of age-related macular degeneration with anti-vascular endothelial growth factor (VEGF) drugs, we often see cases with anti-VEGF-resistant refractory subretinal fluid. In this report, we present two cases of anti-VEGF-resistant refractory age-related macular degeneration (AMD) due to the concurrent development of central serous chorioretinopathy (CSCR) in eyes previously well controlled with intravitreal anti-VEGF injections. **Case presentation**: Two patients underwent intravitreal aflibercept for the treatment of neovascular AMD. Initially, both patients responded well to intravitreal aflibercept, resulting in the complete resolution of the subretinal fluid. However, both patients subsequently developed sudden-onset refractory subretinal fluid that did not respond to repeated intravitreal aflibercept. Fluorescein angiography, indocyanine green angiography, and swept-source optical coherence tomography revealed focal leakage spots, choroidal hyperpermeability, and dilated choroidal vessels, respectively, which were distinct from the pre-existing choroidal neovascularization and suggestive of newly developed CSCR. Laser photocoagulation of the leak spots resulted in the complete resolution of the once-refractory subretinal fluid and the maintenance of vision. **Conclusions**: Our cases highlight that anti-VEGF-refractory subretinal fluid may occur secondary to concurrent CSCR in patients receiving regular anti-VEGF treatments for AMD. In those patients, treatment for CSCR is effective for controlling subretinal fluid that is unresolved by anti-VEGF treatment.

## 1. Introduction

Neovascular age-related macular degeneration (nAMD) is a leading cause of severe vision loss [1]. The injection of intravitreal anti-vascular endothelial growth factor (VEGF) is the current gold standard for nAMD treatment. Numerous clinical trials have confirmed the efficacy of anti-VEGF injections in resolving subretinal fluid (SRF) and improving or maintaining vision. However, some patients may develop a worsening or diminished therapeutic response to anti-VEGF injections during the course of repeated treatments [2,3,4,5].

In this report, we report two patients with nAMD previously well controlled with intravitreal aflibercept (IVA) who developed the sudden onset of refractory subretinal fluid due to the concurrent development of central serous chorioretinopathy (CSCR). Refractory subretinal fluid with repeated IVA was successfully treated by the laser photocoagulation of the leak spots associated with CSCR.

## 2. Case Presentation

### 2.1. Case 1

A 56-year-old woman presented with fibrovascular pigment epithelial detachment (PED) and SRF associated with nAMD in the right eye (Figure 1). Her best corrected visual acuity (BCVA) was 20/20 OD. The patient underwent three monthly IVAs. The PED reduced in height, and the SRF completely resolved. Vision was maintained at 20/20 OD. However, at eight months, SRF recurred at the macula. Vision decreased to 20/25. An additional IVA was performed; however, the SRF increased further after 1 month. The FA showed no leakage from the choroidal neovascularization (CNV), but a new focal leakage appeared, originating adjacent to the CNV lesion. Indocyanine angiography (ICGA) revealed a new hyperfluorescent spot corresponding to the leakage on the FA. Dilated choroidal vessels were also observed. Swept-source optical coherence tomography (SS-OCT) showed a small serous PED corresponding to the leakage spot. Based on these multimodal imaging findings, CSCR was diagnosed as the cause of recurrence and the worsening of SRF. Laser photocoagulation was applied to the leakage spot, resulting in complete resolution of SRF after 1 month. Vision was maintained at 20/20.

### 2.2. Case 2

A 74-year-old man presented with polypoidal choroidal vasculopathy with choroidal nevus in the right eye (Figure 2). His BCVA was 20/60 OD. ICGA revealed a polypoidal lesion and choroidal vascular hyperpermeability, and SS-OCT revealed SRF at the macula. After three monthly IVAs, SRF completely resolved, and vision improved to 20/50. With continuous bimonthly IVA, the macula remained dry for 10 months. However, in month 11, SRF at the macula recurred. Despite three additional consecutive monthly IVAs, the SRF did not improve, indicating refractoriness to aflibercept. The FA revealed a new leakage spot away from the original polypoidal lesion. ICGA also revealed a new area of hyperfluorescence. Based on these findings, we diagnosed CSCR as the cause of worsening SRF fluid. Laser photocoagulation was applied at the leakage spot, and the SRF completely resolved after 2 months. Vision was maintained at 20/50.

## 3. Discussion

We present two patients with nAMD previously well controlled with anti-VEGF injections who experienced the sudden onset of refractory SRF due to newly developed CSCR. We assumed that refractory SRF was due to CSCR, based on the following findings: (1) a focal leakage spot on FA that was not found initially; (2) choroidal hyperpermeability on ICGA around the leakage spot outside the CNV or polypoidal lesion [6,7]; (3) a small serous PED corresponding to the leakage spot on SS-OCT, which is typical in CSCR; (4) the sudden onset of worsening SRF, despite continuous intravitreal aflibercept treatment; and (5) the complete resolution of SRF after focal laser photocoagulation for the leakage spot. These findings strongly suggest that the refractory SRF was due to the secondary development of CSCR. Our cases highlight the importance of suspecting concurrent CSCR in patients with worsening or diminished therapeutic response to anti-VEGF injections during the course of repeated treatments for nAMD.

The current cases illustrate that CSCR may develop secondary to nAMD and cause anti-VEGF-refractory SRF. By contrast, some patients with AMD have a history of CSCR, indicating that CSCR is also one of the underlying pathologies that predispose the patients to develop nAMD. AMD may also develop in patients with features similar to CSCR, such as pachychoroid pigment epithelimopathy (PPE), even if they do not have a history of SRF [8]. Recently, the term “pachychoroid neovasculopathy (PNV)” has been proposed to describe cases with choroidal neovascularization secondary to pachychoroid spectrum, including CSCR, and eyes with PNV are reported to harbor the potential to develop CSCR [9]. Our case 2 is considered a typical PNV, as the PCV accompanied features of PPE at the baseline visit (i.e., the hyperpermeability of the choroidal vessel, a thick choroid, and the dilation of the choroidal vessel); therefore, this case had already possessed the underlying pathology to develop CSCR. In contrast, our case 1 had no obvious PPE features prior to treatment. Thus, CSCR may develop not only in patients with PNV but also in typical AMD patients without features of PPE. Currently, the standard treatment for typical AMD, PCV, and PNV is anti-VEGF therapy. SRF that reappears after the initiation of anti-VEGF therapy is usually considered associated with recurrence from the CNV or polypoidal lesions. However, if SRF does not improve or worsens unexpectedly during intravitreal anti-VEGF injections, clinicians should suspect the potential development of CSCR. Because SRF due to CSCR is indistinguishable from that associated with the recurrence of nAMD, both FA and ICGA are recommended for prompt and accurate diagnosis, as well as for an appropriate treatment either with laser photocoagulation or with photodynamic therapy, as fluid associated with CSCR may not respond well to anti-VEGF treatment. In the present cases, we performed laser photocoagulation as the treatment for secondary CSCR. We believe that photodynamic therapy may also have been effective. However, since the leak spot was obvious and was far away from the fovea and outside the CNV, we chose laser photocoagulation to minimize the risk of retinal pigment epithelium damage from photodynamic therapy. 

Although anti-VEGF treatment is effective in some patients with CSCR, it is ineffective in most cases. [10] Anti-VEGF therapy may inhibit leakage resulting from CNV; however, it has been considered ineffective for subretinal fluid due to CSC. This is why our cases with SRF due to CSCR were refractory to repeated intravitreal aflibercept. 

## 4. Conclusions

Patients with nAMD previously well controlled with intravitreal anti-VEGF injections may develop sudden-onset refractory SRF due to the development of CSCR. The prompt diagnosis of newly developed CSCR may result in appropriate treatment and consequent favorable visual and anatomic outcomes. Clinicians should be aware that a concurrent development of CSCR may be one of the etiologies associated with a worsening or diminished therapeutic response to anti-VEGF injections during the course of treatments for nAMD.

## Figures and Tables

**Figure 1 pharmaceuticals-16-00089-f001:**
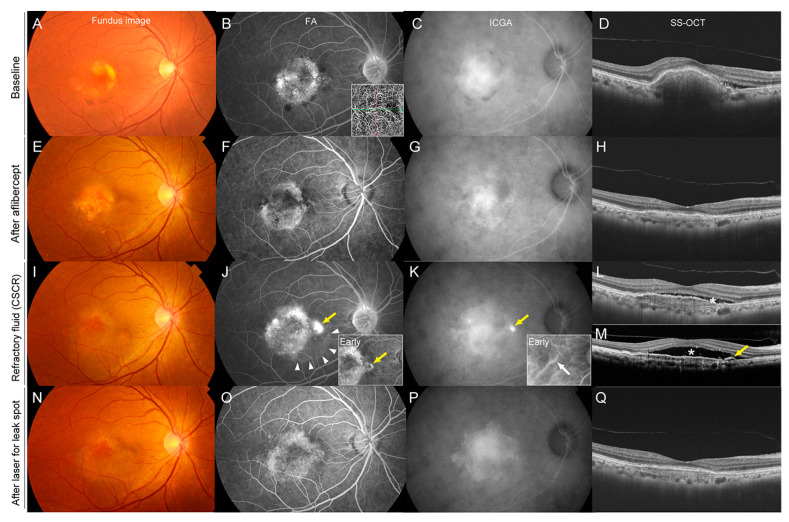
Refractory Age-related Macular Degeneration Due to Concurrent Central Serous Chorioretinopathy in a 56-year-old Woman. Images are at baseline (**A**–**D**), after three consecutive intravitreal aflibercept treatments (**E**–**H**), at the time of the development of refractory subretinal fluid due to CSCR (**I**–**M**), and after laser treatment for leakage spot in CSCR (**N**–**Q**). (**A**) A color fundus photograph at baseline showed choroidal neovascularization (CNV) associated with age-related macular degeneration. (**B**) A late-phase fluorescein angiography (FA) showed leakage associated with CNV, and OCT angiography showed CNV. (**C**) A late-phase indocyanine green angiography (ICGA) showed a hyperfluorescence associated with CNV. (**D**) Swept-source optical coherence tomography (SS-OCT) showed fibrovascular pigment epithelial detachment (PED) and subretinal fluid. (**E**–**H**) After three monthly consecutive intravitreal treatments with aflibercept, the height of the PED reduced and subretinal fluid disappeared. (**I**–**M**) At eight months, the FA showed a newly developed leakage spot adjacent to the CNV margin (**J**, yellow arrow) and the area of serous retinal detachment (white arrowheads). ICGA revealed dilated choroidal vessels in the early phase (white arrow) and a hyperfluorescence corresponding to the leakage in the late phase (**K**). SS-OCT showed recurrence of the subretinal fluid (**L**, asterisk). Despite additional intravitreal aflibercept, subretinal fluid increased further at nine months, indicating refractoriness to intravitreal aflibercept. SS-OCT slice at the leakage spot showed a small serous PED (arrow) corresponding to the leakage spot and underlying dilated choroidal vessels below the leakage spot. (**N**–**Q**) After laser photocoagulation to the leakage spot shown in (**J**), the subretinal fluid completely disappeared (**Q**).

**Figure 2 pharmaceuticals-16-00089-f002:**
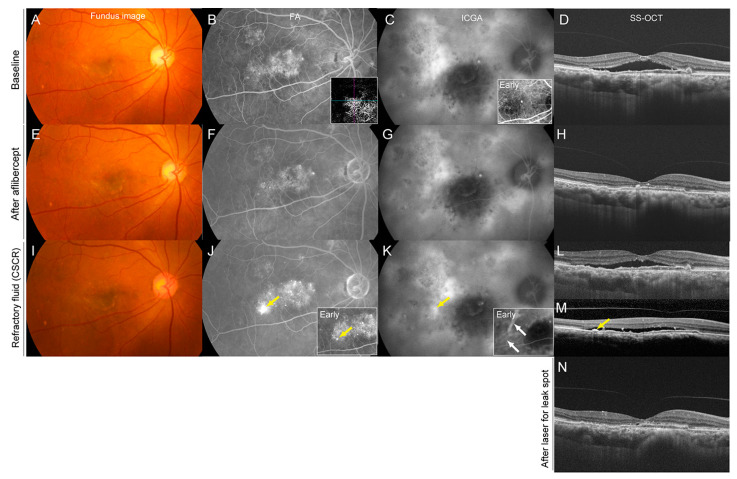
Refractory Subretinal Fluid Due to Concurrent Central Serous Chorioretinopathy in a 74-year-old Man With Polypoidal Choroidal Vasculopathy Accompanied With Choroidal Nevus. Images are at baseline (**A**–**D**), after three consecutive intravitreal aflibercept treatments (**E**–**H**), at the time of the development of refractory subretinal fluid due to CSCR (**I**–**M**), and after laser treatment for leakage spot in CSCR (**N**). (**A**) A color fundus photograph at baseline showed subretinal fluid and the color change associated with a choroidal nevus. (**B**) A late-phase fluorescein angiography (FA) showed leakage from the polypoidal lesion and branching vascular networks, and OCT angiography clearly showed CNV. (**C**) A late-phase indocyanine green angiography (ICGA) showed a polypoidal lesion and branching vascular networks associated with polypoidal choroidal vasculopathy, hyperpermeability of choroidal vessels, and a hypofluorescence due to blockage of the choroidal nevus. (**D**) Swept-source optical coherence tomography (SS-OCT) showed subretinal fluid, a double-layer sign, and thickened choroid. (**E**–**H**) After three monthly consecutive intravitreal aflibercept treatments, the subretinal fluid completely disappeared. (**I**–**M**) At 11 months (2 months after the last intravitreal aflibercept injection), the FA showed a newly developed leakage spot away from the original polypoidal lesion (**J**, yellow arrow). The ICGA revealed dilated choroidal vessels in the early phase (white arrows) and a new hyperfluorescent area in the late phase (**K**). SS-OCT showed recurrence of the subretinal fluid (**L**). Despite three additional intravitreal aflibercept treatments, the subretinal fluid did not resolve, indicating refractoriness to intravitreal aflibercept. SS-OCT slice at the leakage spot showed a small serous PED (**M**, yellow arrow) corresponding to the leakage spot. After laser photocoagulation to the leakage spot shown in (**J**), the subretinal fluid completely disappeared (**N**), although focal ellipsoid disruption remained.

## Data Availability

Not applicable.

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
