# Peer review of "Refractory Age-Related Macular Degeneration Due to Concurrent Central Serous Chorioretinopathy in Previously Well-Controlled Eyes"

_pharmaceuticals, 2023, doi:10.3390/ph16010089_

Round 1
Reviewer 1 Report
Hara et al. presented the two cases with anti-VEGF-refractory SRF, which occurred presumably secondary to concurrent CSCR in patients receiving regular anti-VEGF treatments for AMD.
These case presentations are meaningful and educational. However, I have two suggestions to improve the original manuscript that will need to be addressed before I can fully endorse it for publication.
1)Considering the authors' previous works, this reviewer reckons the authors have taken OCT angiography in these cases; please add the OCT angiography data and add it to the discussion.
2)Please discuss 1: treatment options for CSCR, 2: the mechanism of treatment of anti-VEGF therapy and treatment options for CSCR, and 3:why focal laser photocoagulation was effective in these two cases.
Author Response
Response to Reviewer #1
- Considering the authors' previous works, this reviewer reckons the authors have taken OCT angiography in these cases; please add the OCT angiography data and add it to the discussion.
Thank you very much for your comments and suggestions regarding our paper. We added OCT angiography image to Figure. OCT angiography showed clearly CNV associated with the leakage of FA in both cases.
- Please discuss 1: treatment options for CSCR, 2: the mechanism of treatment of anti-VEGF therapy and treatment options for CSCR, and 3:why focal laser photocoagulation was effective in these two cases.
Thank you very much for your comments and suggestions regarding our paper. We added these discussions to discussion.
Page4 line148-152
In the present cases, we performed laser photocoagulation as the treatment for secondary CSCR. We believe that photodynamic therapy may also have been effective. However, since the leak spot was obvious and was far away from the fovea and outside the CNV, we chose laser photocoagulation to minimize the risk of retinal pigment epithelium damage from photodynamic therapy.
Page4 line 153-156
Although anti-VEGF treatment is effective in some patients with CSCR, it is ineffective in most cases. [10] Anti-VEGF therapy may inhibit leakage resulting from CNV; however, it has been considered ineffective for subretinal fluid due to CSC. This is why our cases with SRF due to CSCR were refractory to repeated intravitreal aflibercept.

Reviewer 2 Report
The author explains the clinical problem of AMD with CSCR through two cases.
Prompt diagnosis of newly developed CSCR may result in appropriate treatment, and consequent favorable visual and anatomic outcomes. Clinicians should be aware that concurrent development of CSCR may be one of the etiologies associated with a worsening or diminished therapeutic response to anti-VEGF injections during the course of treatments for nAMD.
This article has sufficient argument and reasonable summary, which can arouse readers' interest and can be considered for publication
Author Response
Response to Reviewer #2
Thank you very much for your comments. We believe that our cases provide valuable information for patients, ophthalmologists, and retina specialists.
